# The Dynamic Structure and Rapid Evolution of Human Centromeric Satellite DNA

**DOI:** 10.3390/genes14010092

**Published:** 2022-12-28

**Authors:** Glennis A. Logsdon, Evan E. Eichler

**Affiliations:** 1Department of Genome Sciences, University of Washington School of Medicine, Seattle, WA 98195, USA; 2Howard Hughes Medical Institute, University of Washington, Seattle, WA 98195, USA

**Keywords:** centromere, pericentromeric DNA, satellite DNA, genomics, long-read sequencing

## Abstract

The complete sequence of a human genome provided our first comprehensive view of the organization of satellite DNA associated with heterochromatin. We review how our understanding of the genetic architecture and epigenetic properties of human centromeric DNA have advanced as a result. Preliminary studies of human and nonhuman ape centromeres reveal complex, saltatory mutational changes organized around distinct evolutionary layers. Pockets of regional hypomethylation within higher-order α-satellite DNA, termed centromere dip regions, appear to define the site of kinetochore attachment in all human chromosomes, although such epigenetic features can vary even within the same chromosome. Sequence resolution of satellite DNA is providing new insights into centromeric function with potential implications for improving our understanding of human biology and health.

## 1. Introduction

Centromeres are essential chromosomal regions that serve as the site for spindle attachment during mitosis and meiosis and ensure the equal and accurate segregation of chromosomes during cell division. In almost all mammalian species, centromeres are composed of arrays of near-identical tandem repeats, which were identified in early human DNA centrifugation experiments as AT-rich DNA fractions termed “satellite DNA” [1,2,3]. For the past two decades, the majority of centromeric satellite DNA have been excluded from human reference genomes and have been, instead, represented as unfinished sequence gaps or simulated arrays known as “reference models” or “decoys” [4]. In addition, most sequencing-based studies have excluded these regions as part of standard human genetic analyses. Consequently, our understanding of the sequence organization, variation, and evolution of human centromeres has been limited, owing to their highly repetitive nature, large size (often megabases in length), and high sequence identity.

A series of pioneering studies based on pulsed-field gel electrophoresis, Southern blotting, and fluorescence *in situ* hybridizations in the late 1980s and 1990s revealed much about the organization of human centromeric satellite DNA [5,6,7,8,9,10,11]. Human centromeres are mainly composed of six classes of repetitive DNA: α-satellite, β-satellite, ɣ-satellite, and three shorter motifs termed HSATI, HSATII, and HSATIII (Figure 1, Table 1). While α-satellites are found on every chromosome in association with the primary metaphase constriction [12,13,14,15], the other satellites are restricted to a subset of chromosomes [16,17,18,19,20], chromosomal regions [21,22,23,24,25], or secondary constrictions [26,27,28], also called qh regions (Table 1). The kinetochore is largely restricted to higher-order repeat (HOR) units of the tandemly repeating α-satellite, which are flanked on the periphery by more divergent monomeric α-satellite DNA followed by other classes of satellites. While the chromosome-specific α-satellite HORs in humans were all defined by the mid-1990s, only general models existed for how centromeres are organized and have evolved [29], with a limited understanding of their precise sequence composition and how they vary among human and nonhuman primates [30,31,32,33,34,35,36,37,38]. Moreover, the site of kinetochore attachment was not clearly defined, often being inferred based on lower-resolution cytogenetic and immunohistochemical experiments or through limited sequence analyses [39,40,41]. With the application of long-read sequencing technologies, centromeric satellites can now be fully sequenced and assembled [42,43,44,45]. We review the complete sequence of human centromeric satellite DNA with an emphasis on the new insights that have emerged and how our model of centromeric DNA has been further refined, with potential implications for improving our understanding of human biology and health.

## 2. A Complete Census of Centromeric Satellite DNA from One Human Genome

The Telomere-to-Telomere (T2T) Consortium recently resolved the first complete sequence of a human genome (T2T-CHM13) [45], and, in doing so, unveiled the sequence composition of all human centromeres (Figure 2) [44]. There were three advances that made this possible: (1) the use of a complete hydatidiform mole (CHM), where no allelic variation existed to confound assembly of highly repetitive and identical haplotypes; (2) the use of Pacific Biosciences (PacBio) high-fidelity (HiFi) sequence data, which generated a highly accurate sequence backbone of nearly all of the human genome; and (3) the application of ultra-long Oxford Nanopore Technologies (UL-ONT) sequence data, which allowed sequence contigs to be effectively scaffolded. The latter was especially critical to traverse the megabases of repetitive DNA constituting the centromeric regions of the human genome [42,43,44]. Accompanying these advances in technology were a series of rapid-fire-in-succession genome assembly algorithms, such as HiCanu [46], hifiasm [47], and Verkko [48] that leveraged the unique attributes of the different long-read sequencing technologies or specialized in the characterization and validation of centromeric satellite DNA assemblies [49,50,51]. It should be noted that subsequent development of methods to accurately phase diploid genomes [52,53,54] has meant that use of a CHM is no longer necessary, and centromeres from diploid human genomes have now been readily assembled [42,44,48]. However, centromeric regions are still preferentially associated with breaks in standard long-read sequence assemblies, such as in the Human Pangenome Reference [55].

Centromeres and their associated pericentromeric DNA are estimated to constitute ~6.2% (189 Mbp) [44] or 7.1% (221.7 Mbp, if the Y chromosome is included) [57] of the human genome and are largely composed of megabases of α-satellite, β-satellite, ɣ-satellite, and three human satellites (HSATI, HSATII, HSATIII) distributed differentially among human chromosomes (Figure 1, Table 1). It should be noted that there is no cytogenetically recognized pericentromere in humans, but the term “pericentromeric DNA” was originally used to describe the five Mbp of DNA extending on either side of the higher-order α-satellite [36]. In humans, pericentromeric DNA contains various classes of inactive α-satellite, almost all other forms of satellites, and large blocks of recent segmental duplication shared among non-homologous chromosomes [36]. It represents the transition region to euchromatin. Later, more nuanced approaches refined this definition to represent the haplotypes flanking the centromere, termed “cenhaps” [58], which tend to evolve as a single chromosomal segment due to infrequent recombination and extensive linkage disequilibrium [44]. In humans, α-satellite, a ~171 bp repeat, is the most abundant, spanning 85.7 Mbp (2.8%) of the human genome and almost exclusively associated with the kinetochore [44,45,57]. Most α-satellite DNA are organized into higher-order arrays consisting of discrete units of monomers repeated in tandem hundreds to thousands of times and flanked on their periphery by divergent HORs and monomeric α-satellite. While most human chromosomes harbor more than one related α-satellite HOR array, only one HOR array is typically associated with the kinetochore, and these are defined as “active” α-satellite HOR arrays. Active α-satellite HOR arrays were first identified in studies of dicentric chromosomes, which have one “active” and one “inactive” centromere. Using immunofluorescence microscopy, these studies revealed that active centromeres were enriched with nucleosomes containing the histone H3 variant CENP-A, while the inactive centromeres were not [59,60]. In general, α-satellite DNA extend contiguously with occasional interruption by mobile element retrotransposition events or blocks of other satellite DNA (for example, the *D3Z1* and *D4Z1* α-satellite HOR arrays on chromosomes 3 and 4, which are disrupted by an array of HSATI repeats). Sequence analysis reveals few examples of inversions within the HOR units, suggesting the orientation of α-satellite is typically maintained [44].

Human centromeric α-satellites are categorized into twenty different suprachromosomal families (SFs; SF1-18, SF01, and SF02), which are groups of α-satellites that are more closely related to each other than to other groups [31,44,61,62,63,64]. The sequence identity between α-satellite HORs in an SF is ~85–88%, and between different SFs, it is 50–85% [44]. The three main SFs, SF1-3, represent the “active” (kinetochore-binding) α-satellite HOR arrays on all human autosomes and the X chromosome, and they are composed of either dimeric (SF1 and SF2) or pentameric (SF3) monomer configurations [63]. The SF4 and SF5 families usually flank the active α-satellite HOR arrays and are composed of either purely monomeric repeats (SF4) or a combination of monomeric and divergent HORs that lack a regular repeat structure (SF5) [30,32,63]. Thirteen minor SFs (SF6-18) represent older, more ancient α-satellite monomers that either reside on the extreme edges of centromeric regions or are located far away from the centromere, potential relics of long-defunct ancient centromeres [44]. Finally, SF01 and SF02 are recently defined SF classifications representing mixtures of SFs residing in pericentromeric DNA [44,64,65].

The transition to euchromatin, based on CpG methylation profiling for most chromosomes, is relatively sharp [42,43,44,66], with the exception of the acrocentric chromosomes (chromosomes 13, 14, 15, 21, and 22), where blocks of α-satellite extend into the short arms of the chromosomes, interdigitating among blocks of segmental duplications (SDs) and rDNA clusters (Figure 2). While SDs and other classes of satellite DNA map peripherally to α-satellite centromere-associated DNA, relatively few functional protein-coding genes have been identified within pericentromeric DNA, consistent with detailed analyses of these regions two decades earlier, which predicted an abundance of unprocessed pseudogenes [36,67]. Of the 676 potential gene annotations, only 23 correspond to validated protein-coding genes, such as *KCNJ17* and *UBBP4*. One of the most remarkable features of pericentromeric DNA is the extent of interchromosomal homology among subsets of human chromosomes (Figure 3). An analysis of various classes of satellite DNA provided compelling evidence that β-satellite, HSATI, and HSATIII share the highest degree of homology among the acrocentric chromosomes (Figure 3) [44]. Similarly, pericentromeric segmental duplications are the largest and most identical among the short arms of chromosomes 13, 14, 15, and 22, with the intervals between the centromeric satellite and secondary constrictions (qh regions) on chromosomes 1, 9, and 16 showing some of the highest degree of interchromosomal homology (Figure 3) [68]. While cause and consequence are difficult to disentangle, it is likely that this homology facilitates association of acrocentric chromosomes to form the nucleolus as well as ectopic exchanges and duplication among specific subsets of nonhomologous chromosomes.

Consistent with original theoretical expectations and subsequent phylogenetic analyses [29,32,34], most centromeric α-satellite HORs are organized into evolutionary layers, with divergent α-satellites residing on the periphery and becoming increasingly more homogenous and displaying high sequence identity within the interior [42,44]. Layers are distinguished from more gradual decay or divergence of HORs by relatively sharp transitions of sequence homology (see numbered arrows for Figure 4a). For some chromosomes, the organization appears highly symmetrical, creating a mirror-like perspective as first noted for chromosome 8 [42] and subsequently observed for chromosomes 17, 18, and 19 [44]. For other centromeres, the α-satellite HOR array is more homogeneously or asymmetrically distributed, although it should be stressed that most of our current understanding has been shaped by the analysis of one human genome. Given the amount of variation observed in sequence and structure among human haplotypes, many more complete centromeres will need to be surveyed. New analytical (e.g., NTPrism [44], HORmon [69], and CentromereArchitect [70]) and visualization (e.g., StainedGlass [56]) tools that were developed to facilitate the identification of these even higher-order patterns within α-satellite and other satellite DNA will be important for future studies of satellite DNA variation and evolution.

## 3. CpG Methylation and the Discovery of the DNA Site of Kinetochore Attachment

In addition to facilitating the assembly of centromeric regions, long-read sequencing also allowed direct detection of CpG-methylated base pairs from native DNA. Initial studies of chromosome 8 and X centromeres both observed a conspicuous region of hypomethylation (approximately 61–73 kbp in length) buried within the hypermethylated active α-satellite HOR array [42,43]. Using chromatin immunoprecipitation followed by sequencing (ChIP-seq) experiments, Logsdon was the first to show that this region was enriched for the centromeric histone CENP-A—an observation validated by CENP-A immunostaining on chromatin fibers [42] (Figure 4). In the case of chromosome 8, CENP-A enrichment extended over a broader stretch of 632 kbp, but the peak enrichment centered over the hypomethylated α-satellite HORs. Because CENP-A is a histone H3 variant specifically associated with the centromere [71], these observations suggested that the hypomethylated pocket represents the binding site of the functional kinetochore. The broader CENP-A chromatin peak may represent variability in the position of CENP-A nucleosomes among a population of cells, which has been observed on individual chromatin fibers from native centromeres [72]. Subsequent follow-up experiments, including CUT&RUN, confirmed these general properties on the remainder of the centromeres within the T2T-CHM13 genome [44,66]. The conspicuous hypomethylated region on each centromere was later termed the “centromere dip region” or CDR [44,66]. Genome-wide analyses showed that CDRs were typically constrained to 26–423 kbp in length with CENP-A enrichment extending 190–570 kbp within the active α-satellite HOR array [44]. CDRs mapped only to the active α-satellite HOR arrays, which are typically among the largest and show the highest degree of CpG methylation [66]. In many cases, the CDR and CENP-A enrichment was associated with the evolutionarily youngest and recently expanded α-satellite HOR array, although this was not a universal observation for all centromeres [44]. In the case of chromosome 8, both the CDR and the CENP-A-enriched region mapped to a more diverse set of α-satellite HORs (Figure 4). Similar offsets with respect to the most recently expanded α-satellite HORs were observed for chromosomes 5, 7, and 13. Because the site of kinetochore attachment and hypomethylation are epigenetic properties, caution should be exercised in drawing genetic correlations with the composition of the α-satellite HORs until more genomes and tissues have been examined. Preliminary analyses suggest that the site of kinetochore may, in fact, vary considerably depending on the haplotype in question.

## 4. Variation in Centromeric Satellite Sequence and Structure

Centromeric satellites are prone to single-nucleotide and structural variation induced by mutational processes such as unequal crossover [29] and gene conversion [73]. These processes can result in rapid changes in satellite sequence composition, repeat structure, and copy number. Early studies using cytogenetic and gel-based techniques revealed that centromeric satellites often undergo repeat amplifications and contractions, which can result in dramatic changes in satellite array size on the order of tens to thousands of kilobases [74,75]. While the detection of large-scale variation in satellite array structure is feasible with cytogenetic and gel-based techniques, discovery of finer-scale variation, such as changes in satellite sequence composition or repeat structure, has been historically difficult to achieve. With complete sequence assemblies of centromeres from multiple human genomes [42,44,76], however, detection of these more fine-scale variants has recently become attainable. A comparison of the chromosome Y *DYZ3* α-satellite HOR array from 21 diverse human genomes using high-quality sequence assemblies, for example, recently enabled the discovery of a 36-mer α-satellite HOR in 52.4% of human haplotypes [76] (Figure 5a). This α-satellite HOR is thought to be an ancestral version of the canonical 34-mer α-satellite HOR, which was born out of repeated deletions of α-satellite monomers at the 22nd monomer position. Similarly, a comparison of the chromosome X *DXZ1* α-satellite HOR array from seven diverse human genomes revealed the presence of duplications spanning hundreds of kilobases in two human genomes (HG01109 and HG03492) as well as the emergence of an α-satellite HOR haplotype predominantly in those with African ancestry [44] (Figure 5b). Finally, pairwise sequence comparisons of the chromosome 8 centromeric region from three human haplotypes revealed gross variation in the sequence and structure of the *D8Z2* α-satellite HOR array, with the pairwise single-nucleotide variant sequence identity dropping down to 99.6% on average, significantly lower than that of the pericentromeric flanking sequences [42] (Figure 5c). As more and more human genomes are sequenced and assembled, the catalog of novel α-satellite sequence and structural variants will become more complete, facilitating more sophisticated models of human centromeric satellite variation. Such models should ultimately distinguish both hypervariable and conserved structural features relevant to chromosome segregation and disease.

## 5. Human Centromere Evolution

Numerous comparative sequence studies between human and nonhuman genomes have shown that centromeric satellites evolve at an accelerated pace due to the action of mutational processes, including concerted gene evolution, saltatory amplification, and unequal crossover [30,32,33]. Estimating the actual increase in mutation rate, however, remains challenging due to the difficulty of sequence alignment [49], even when orthologous centromeres are completely sequenced among closely related species. Comparing the human and chimpanzee chromosome 8 centromeres, for example, the α-satellite HOR array is too divergent to generate a simple pairwise alignment that would permit the computation of a mutation rate over the last six million years since speciation [42] (Figure 6a). Reliable one-to-one alignments, however, have been made spanning the α-satellite monomeric transition regions and, even in these more tractable portions, we find evidence of increased allelic divergence of at least 2.2-fold. Phylogenetic reconstructions focused on the 171 bp α-satellite monomer itself clearly show that the peripheral α-satellite repeats evolve more slowly than the α-satellite HORs, revealing phylogenetic relationships between macaques and humans (~25 million years ago) and defining potential ancestral centromere regions shared among these diverse lineages [32,42]. The analysis also reveals distinct evolutionary trajectories for the emergence of both the dimeric α-satellite (predominant among Old World monkeys) and the higher-order α-satellite (common to the great ape lineages). The evolutionary turnover of α-satellite HORs is extraordinary as novel α-satellite repeats emerge, amplify, and homogenize through mechanisms such as unequal crossover and gene conversion—a common feature among many species [77]. Different lineages appear to have different characteristics with respect to higher levels of organization; among orangutans, for example, composite HORs have been noted where the HOR layers show relatively little sequence homology among themselves (Figure 6b), in contrast to Old World monkeys, where dimeric repeat structures are distributed among all autosomes [42]. This rapid and stereotypic evolution of centromeres has been described as a potential driving force for speciation, due to the accumulation of sequence differences that result in highly divergent α-satellite HOR sequences and, subsequently, cause incompatibility and reproductive barrier in hybrids between closely related species [78,79]. Even within a species, however, there is extraordinary single-nucleotide and structural variant diversity, possibly as a result of ongoing centromeric meiotic drive to segregate more efficiently [80]. Without this selection, centromeric satellites degrade very quickly, as is evidenced by the structure of the inactivated human chromosome 2 centromere, which reduced from 4.04 Mbp of α-satellite HORs to 41 kbp of divergent α-satellite monomers after the chromosome 2p/2q fusion occurred in the ancestral human lineage (Figure 6c). In this light, it is interesting that sequence comparisons among three human centromere 8 haplotypes show regions of excess allelic variation and structural divergence, although the location and composition of HORs differ among haplotypes [42] (Figure 5c). It is likely that evolutionary reconstructions among different species will be preceded by first reconstructing the dynamic mutational changes that distinguish major haplotypes within species and relating these changes to relocations of kinetochore attachment. Such analyses should, in turn, lead to improved assembly algorithms and improvements in mutational modeling of such complex regions.

## 6. Future Directions

The complete sequence of a human genome has significantly advanced our understanding of the sequence composition, organization, DNA methylation patterns, and chromatin landscape of satellite DNA, but these findings only reflect that of a single human genome. The dynamic nature (Figure 5) and rapid evolution (Figure 6) of human satellite DNA suggest that these sequences are likely to be among the most variable among humans. As such, one representation of satellite sequence and structure is far from sufficient, and many more genomes will need to be sequenced to accurately model genetic variation in these regions. Efforts to assess the variation of satellite sequences among the human population are currently underway, with the Human Pangenome Reference Consortium (HPRC) expected to sequence at least 350 diverse human genomes [81] and the T2T Consortium proposing to finish a subset of these. Given the considerable single-nucleotide and structural variation already observed among a subset of human centromeric haplotypes [42,44,46], it is likely that many more genomes will need to be sequenced and assembled for the genetic diversity of these regions to be sufficiently represented and understood. Accurately representing the complex forms of genetic variation in a graph-based pangenome reference, however, is an unmet challenge requiring significant algorithmic development [82,83,84,85]. Other efforts to generate complete and accurate assemblies of nonhuman primates are also ongoing, which will provide the necessary outgroups to reconstruct the evolutionary history of these and other highly dynamic regions. As the cost for long-read sequencing continues to plummet and a USD 1000-long-read-sequenced genome comes within reach [86], we anticipate that the generation of nearly complete, phased human genome assemblies will also become routine. This development is especially important for individuals who have rare or complex diseases with no known genetic or epigenetic cause, who stand to benefit from complete sequence resolution of satellite DNA and other previously unresolved regions of their human genome.

The availability of hundreds of thousands of genomes from both healthy and diseased individuals will also advance our understanding of centromere biology. In particular, it will allow one to delineate the genetic relationship between satellite DNA variation and the site of kinetochore attachment, if one exists. It is possible and even likely that variation in the α-satellite HOR array structure and/or chromatin landscape affects the accuracy of chromosome segregation during cell division, and such differences may contribute to infertility, trisomy disorders, and aneuploidy associated with cancer. Detection of pathogenic variants within these regions, whether genetic or epigenetic, will also benefit from combining long-read sequencing data with electronic health records from thousands of individuals to discover significant associations with disease. Federally funded efforts, such as the NIH *All of Us* Research Program, which aims to make de-identified genomic sequencing data and medical records available, has recently embarked on a long-read sequencing initiative to generate data from more than 10,000 genomes [87]. Such efforts promise to accelerate research into these more complex regions of the genome and will serve as a model for other biobank efforts in the future. Critical to this is the continued commitment that genomic data, once de-identified, should become publicly available in order to advance both basic and translational biomedical research. This approach, which was initiated and widely accepted in the early days of the Human Genome Project, has benefitted the greater scientific research communities and will continue to accelerate and democratize genomic research as we begin to access some of the most genetically complex and dynamic regions of our genomes.

## Figures and Tables

**Figure 1 genes-14-00092-f001:**
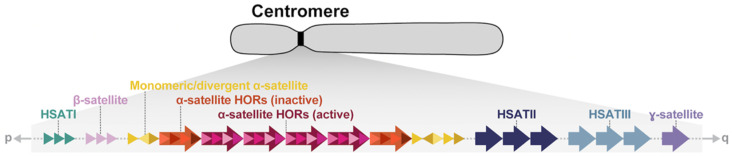
**Model of human centromeric and pericentromeric regions.** Schematic of the generalized organization of human centromeres and their flanking sequence. Major components and their structures are shown. HORs, higher-order repeats; HSAT, human satellite. Not drawn to scale.

**Figure 2 genes-14-00092-f002:**
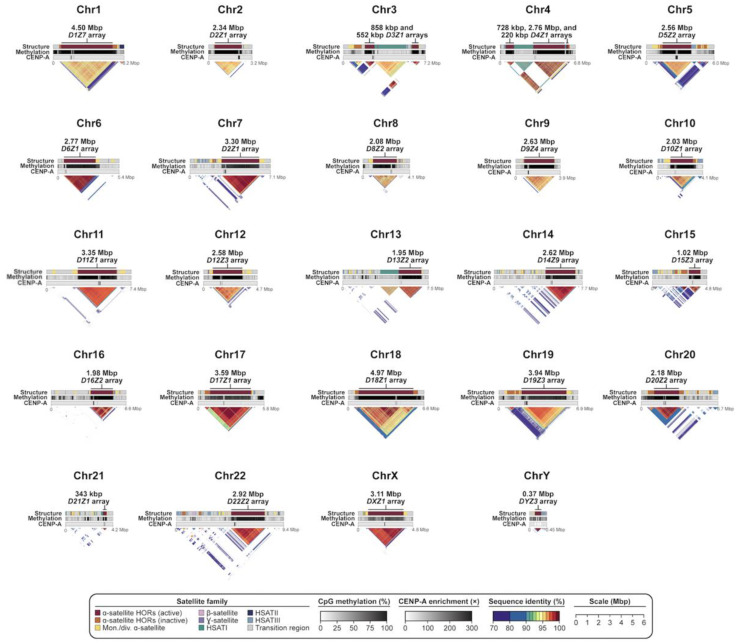
**Sequence composition, DNA methylation pattern, and CENP-A chromatin organization of each centromere in the T2T-CHM13 genome.** Tracks showing the sequence composition, frequency of methylated CpG dinucleotides, and fold-enrichment of CENP-A ChIP-seq reads over bulk nucleosomal reads (or in the case of chromosome Y, the number of CENP-A CUT and RUN reads) for each centromere in the T2T-CHM13 v2.0 genome. Triangular StainedGlass [56] plots indicate the pairwise sequence identity between 5 kbp segments along each centromeric region and are colored by sequence identity. Warmer colors indicate higher sequence identity, and colder colors indicate lower sequence identity (as indicated in the legend).

**Figure 3 genes-14-00092-f003:**
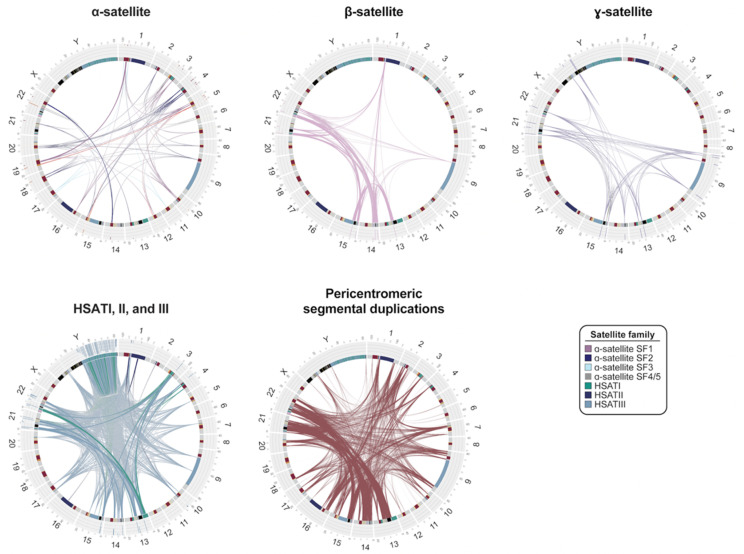
**Interchromosomal relationships between centromeric satellite DNA and pericentromeric segmental duplications.** Circos plots showing sequence relationships among four different satellite families as well as pericentromeric segmental duplications in the T2T-CHM13 genome. Connecting line widths for satellite families indicate the proportion of 75-mers shared between arrays (i.e., thicker lines indicate greater overall sequence similarity between different arrays of the same family). α-satellite lines are colored by their suprachromosomal family (SF) assignment. Radial bar plots indicate specificity of 75-mers proportionally, with white indicating 75-mers unique to the array, light gray indicating 75-mers shared with other centromeric regions, and black indicating 75-mers shared with regions outside of centromeres. The pericentromeric segmental duplication circos plot shows pairwise alignments between pericentromeric regions that are >1 kbp and >90% identical.

**Figure 4 genes-14-00092-f004:**
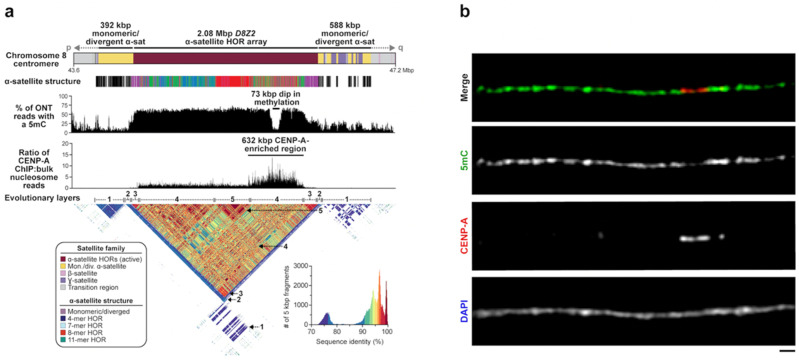
**The site of kinetochore attachment within the chromosome 8 centromere.** (**a**) Schematic showing the sequence composition, α-satellite structure, CpG methylation frequency, and CENP-A chromatin organization of the chromosome 8 centromere in the T2T-CHM13 genome. The *D8Z2* α-satellite HOR array is 2.08 Mbp long and is generally methylated, except for a 73 kbp region enriched with nucleosomes containing the histone H3 variant CENP-A. CENP-A chromatin resides on structurally diverse α-satellite HORs. (**b**) Representative images of a CHM13 chromatin fiber showing CENP-A enrichment in a hypomethylated region. Scale bar, 1 μm. Adapted from [42].

**Figure 5 genes-14-00092-f005:**
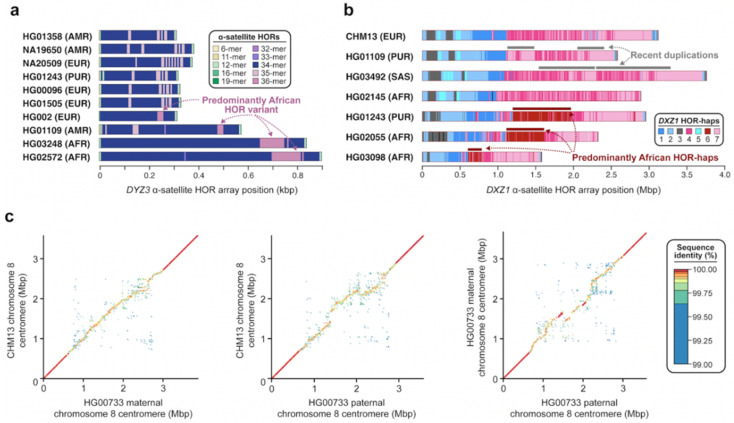
**Sequence and structural variation within centromeric α-satellite HOR arrays.** (**a**) Plots showing the structure of the chromosome Y *DYZ3* α-satellite HOR array in ten diverse human genomes, highlighting the presence of a 36-mer α-satellite HOR variant in four haplotypes (HG002, HG01109, HG03248, and HG02572) [76]. (**b**) Plots showing the α-satellite HOR haplotypes (HOR-haps) present in the chromosome X *DXZ2* α-satellite HOR array in seven diverse human genomes. Two genomes (HG01109 and HG03492) harbor a haplotype with a recent duplication, while three others (HG01243, HG02055, and HG03098) harbor a haplotype that is especially prevalent among those with African ancestry. Adapted from [44]. (**c**) Plots showing the pairwise sequence identity between chromosome 8 centromeric regions from three human haplotypes (CHM13 and two haplotypes from HG00733). The *D8Z2* α-satellite HOR array shows variation in sequence and structure, while the flanking sequences do not. Adapted from [42].

**Figure 6 genes-14-00092-f006:**
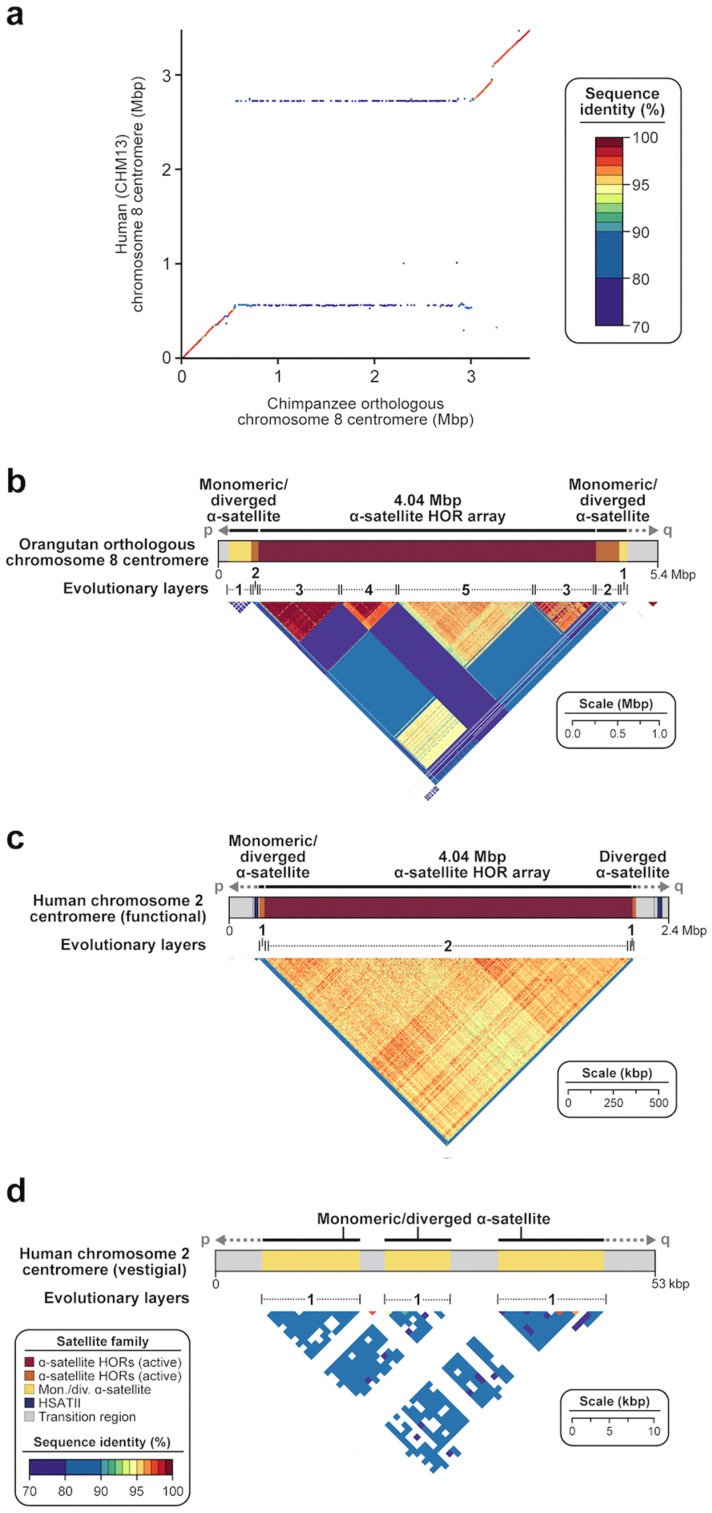
**Evolution of centromeric satellite regions.** (**a**) Dot matrix plot showing pairwise sequence identity between human (CHM13) and chimpanzee chromosome 8 centromeric regions. While there is >95% sequence identity in the monomeric α-satellite sequences and 70–80% sequence identity in the transition regions between monomeric α-satellite and α-satellite HORs, there is almost no sequence identity shared between sequences in the α-satellite HOR array. Despite being similar in size, the two apes show nearly complete turnover of their α-satellite HORs, with best matches occurring at the periphery (horizontal lines). (**b**) StainedGlass view [56] showing the organization and pairwise sequence identity among 5 kbp segments across an orangutan orthologous chromosome 8 centromere. Orangutan α-satellite HOR arrays are composed of mosaic blocks of α-satellite HORs with <95% sequence identity shared between them. Adapted from [42]. (**c**,**d**) StainedGlass view [56] showing the organization and pairwise sequence identity among 5 kbp segments across (**c**) the functional human (CHM13) chromosome 2 centromere and (**d**) the vestigial human (CHM13) chromosome 2 centromere. The vestigial centromere is a remnant of the chromosome 2B centromere present in chimpanzee. It is highly degraded and reduced in size with little obvious α-satellite HOR structure, owing to its inactivity.

**Table 1 genes-14-00092-t001:** Composition and abundance of repeats within human centromeric and pericentromeric regions.

Peri/Centromere Repeat Type	Repeat Unit Length (bp)	GC Content (%)	Abundance inthe T2T-CHM13 Genome (Mbp; %)	Chromosomal Location in the T2T-CHM13 Genome																							
				1	2	3	4	5	6	7	8	9	10	11	12	13	14	15	16	17	18	19	20	21	22	X	Y
α-satellite	~171	39	85.67; 2.75																								
β-satellite	68	52	8.61; 0.28																								
ɣ-satellite	~217	59	0.65; 0.02																								
HSATI	42 (1A)	22 (1A)	13.39; 0.43 (1A)																								
	2420 (1B)	24 (1B)	15.32; 0.49 (1B)																								
HSATII	Variable	37	28.71; 0.92																								
HSATIII	Variable	41	69.33; 2.22																								

Black square: >1 Mbp of sequence is present on the indicated chromosome; dark gray square: 100 kbp–1Mbp is present; light gray square: 10–100 kbp is present; white square: <10 kbp is present. Calculations include satellites from chromosomes 1–22, X, and Y in the T2T-CHM13 v2.0 genome.

## Data Availability

The sequence of the T2T-CHM13 v2.0 genome is available at GenBank accession GCA_009914755.4, and the sequence of the human and nonhuman primate chromosome 8 centromere assemblies are available at GenBank accessions OM502025-OM502028. The Human Pangenome Reference Consortium (HPRC) whole-genome long-read sequence assemblies are available at https://github.com/human-pangenomics/HPP_Year1_Assemblies (accessed on 14 July 2021), and the Human Genome Structural Variation Consortium (HGSVC) whole-genome long-read sequence assemblies are available at http://ftp.1000genomes.ebi.ac.uk/vol1/ftp/data_collections/HGSVC2/release/v1.0/assemblies/ (accessed on 5 June 2021). The data and code used to generate the circos plots are available at the following GitHub repositories: https://github.com/msauria/T2T_Kmer_Analysis (for satellites; accessed on 9 December 2022) and https://github.com/mrvollger/Data-Analysis-for-SDs-in-T2T-CHM13 (for segmental duplications; accessed on 9 December 2022).

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
