# Peer review of "The Dynamic Structure and Rapid Evolution of Human Centromeric Satellite DNA"

_genes, 2022, doi:10.3390/genes14010092_

Round 1
Reviewer 1 Report
This is a well-written review article with specific focus on the centromeric satellite DNA structure. It takes advantage of the recent advances in long-read sequencing technology that allow accurate sequencing of long repetitive genomic region at centromere.
On of the potential limitation for this manuscript is the limited availability of sequencing data from a diverse population, which may limit the generalization of the featured that are presented in this review. However, this is acknowledged by the authors.
Reviewer 2 Report
This review outlines and discusses the significance of key highlights that have emerged from the recent completion of human centromere sequence composition by the T2T Consortium. This excellent and cutting edge review is insightful, well organised, clearly written and extremely relevant for the field. It is accessible even for non-specialists and is relevant for all biologists with an interest in genomes and repeated sequences.
The cited references are from very recent publications, many in 2021 or 2022, which reflects the rapid movement of the human centromere field since T2T completion. The authors also give full credit to critical experiments carried out in the preceding 40 years of research that have led to this point. Importantly, the authors are cautious in drawing general conclusions and emphasise that detailed data so far has been generated from a single human genome. They clearly state that complete sequencing of additional genomes is key understating the full significance and applicability of current observations. Future directions outlined are exciting and far-reaching. The figures and tables are appropriate and are well explained in the legend, supported by the text.
Minor comments and suggestions:
Related to the Introduction and Table 1 and P3:
Clarification of sequences considered to comprise the pericentromere would be useful.
Table 1: It is not clear why ‘⍺-satellite, ~171, ‘39%’ are shown in in bold? There appears the be a double line that is cut short in this panel also.
P3, Line 92: The authors could provide a little more detail on how active HOR arrays were identified experientially.
P6, section on CpG methylation dip: The authors could speculate as to why the CENP-A enrichment extends over a broader region than the CDR.
Although addressed in itself by the Human Pangenome Project, it would be nice to include a brief statement of the importance of diversity in human genome sequencing.
